# Case Study Demonstration of the Potential Acceptability and Effectiveness of a Novel Telehealth Treatment for People Experiencing Gambling Harm

**DOI:** 10.3390/ijerph192316273

**Published:** 2022-12-05

**Authors:** Jane Oakes, Vicky Northe, Chris Darwin, Liza Hopkins

**Affiliations:** 1Alfred Health, Wellbeing and Recovery Research Institute, Adelaide, SA 5000, Australia; 2Wellbeing and Recovery Research Institute, Adelaide, SA 5000, Australia

**Keywords:** gambling harm, group treatment, telehealth, exposure therapy, treatment barriers

## Abstract

The evidence base for internet therapies is building but little is known yet about the acceptability and effectiveness of providing telehealth online in a group format for the treatment of gambling disorders. Therefore, this uncontrolled, real-world study aimed to evaluate the feasibility and effectiveness of providing evidence-based treatment in a group format using an online platform. This innovative approach to treatment of people experiencing gambling harm was developed during the COVID pandemic so that gamblers could access evidence-based treatment from their homes. A closed group treatment program was developed using telehealth, enabling gamblers to come together weekly to engage in a treatment program based on behavioral therapy using cue exposure. Four online gamblers who met the criteria for Gambling Disorder were recruited from a gambling help service. A case report evaluation methodology was used to gain an in-depth understanding of the effectiveness of this approach to treatment. Treatment was conducted weekly over 12 months. All participants engaged with the program and after completing treatment participants no longer met the criteria for a Gambling Disorder, were abstinent for 12 months post treatment and achieved improved life functioning. This program provides preliminary evidence that providing treatment online in a group setting can be an acceptable and effective model in the delivery of treatment for clients unable to attend face-to-face clinics or preferring telemedicine as an option for treatment delivery. These findings warrant further exploration through a larger randomized controlled study.

## 1. Introduction

Gambling in Australia is a significant cause of harm to individuals, families, communities, and society. It has been estimated that 1.1 million regular gamblers behave in ways that cause or put them at risk of gambling-related problems [1]. These gamblers spend much more of their households’ income on gambling than other regular gamblers. Gamblers experiencing severe problems in low-income households spend an average of 27% of their disposable household income on gambling [1]. A conceptual framework for gambling-related harm has identified seven domains related to gambling harm, which include: financial, relationships, emotional and psychological, decrements in health, reduced performance at work and study, cultural harm, and criminal activities [2].

In addition to these harms, there is a well-established correlation between gambling and other comorbidities such as addiction and mental health issues. Common co-morbid mental health disorders include alcohol and other drug use disorders, mood, anxiety, impulse control, and personality disorders. The contribution of gambling itself is difficult to isolate and measure, and causal pathways may likely work in both directions, with problem gambling leading to mental health conditions such as anxiety and depression, while pre-existing conditions such as impulse control disorders may exacerbate gambling behaviours [3]. In addition, there is a significant correlation between gambling problems and suicidal ideation, as well as death by suicide [4]. Gambling-related harms such as interpersonal losses and conflicts, including relationship breakdowns, increase the likelihood of a suicide attempt [5]. Research highlights that clients with a history of suicide attempts are at the highest risk of future suicidal behaviour, including death by suicide [6,7]. A study conducted in Melbourne found that approximately 17.6% of people presenting to local crisis or emergency mental health services were also experiencing problem gambling behaviours [8]. This study led to the development of a state-wide Gambling Harm and Mental Health Service now called Gambling Minds where this treatment program was conducted.

During COVID-19, although gamblers experienced limited access to venues, in general gamblers gambled more often due to the increases in the frequency of gambling online on racing (horse, greyhound, and harness), sports, eSports, lotto, and casino table games. It was reported that almost 1 in 3 gamblers signed up for a new online betting account during COVID-19, and 1 in 20 started gambling online with an increase in their frequency of gambling and an increase in monthly spending on gambling from $687 to $1075 [9].

As part of COVID-19 lockdown restrictions, many health services were affected, limiting people’s access to these services [10], so changes to health service delivery models were essential to ensure that gambling problems continued to be treated effectively and the harm from gambling was reduced. Fortunately, the provision of videoconferencing in psychotherapy has been shown to have promising outcomes with a high client-rated therapeutic alliance. In addition, clients report the enhanced control and personal space provided by video therapy can enhance their confidence to interact and openly discuss feelings and problems [11]. In addition, internet treatment programs are cost-effective and can combine many components of successful cognitive behavioural therapy in a dynamic and interactive format tailored to individual users to increase relevance and effectiveness [12].

The present project was developed to provide effective and flexible treatment to those people experiencing gambling harm at a time when accessing these treatments was hampered by lockdowns and social isolation requirements. To meet this objective, the Gambling Minds service in Melbourne, Australia conducted a small, evidence-based Telehealth Treatment group for people affected by Gambling Harm. The aim of this study was to determine the feasibility and effectiveness of providing Cognitive Behaviour Therapy (CBT), with a specific focus on graded exposure and response prevention, in an online closed group program for people with problem gambling behaviour.

## 2. Method

### 2.1. Design

A case study design was used to report on four clients recruited from gambling help services in Victoria, Australia after receiving financial and therapeutic counselling but not progressing despite many years of support. The case study approach is suited to the exploration of innovative treatments with small numbers of participants in a real-world setting [13]. Standard quantitative measures were used to assess the effectiveness of the online treatment. Problem gambling severity and perceived problem and goal difficulty was measured at treatment commencement, mid-point, and completion. Well-being was measured monthly until treatment completion. Qualitative research can help provide a broader understanding of clinical realities [14], therefore a survey including open-ended questions was also completed at 12 months post-commencement to elicit a rich understanding of the client’s treatment experience. 

### 2.2. Participants

The participants referred to the program were four Australian males aged between 39 and 51 years. Each client met the criteria for Gambling Disorder, which the treating clinician diagnosed using a structured interview and the Problem Gambling Severity Index (PGSI) [15]. At the commencement of treatment, all four clients were receiving supportive counselling from a government gambling help service. Once engaged in the group program, these clients subsequently decided to discontinue the ongoing counselling support for the treatment offered in this study. All four clients who took part in the program had at least one co-morbid mental health condition at assessment, including major depression and a history of recent alcohol use disorder. The following provides a summary of the four cases.

#### 2.2.1. Case 1

A 48-year-old married man with a supportive wife who was aware of his gambling-related issues. He has two adult children who live with both parents. He lives with his family at his wife’s parents’ house after selling the family home due to gambling debts. He worked as a CEO for a large company. He presented with a 30-year gambling history of sports betting both online and at off-course betting shops, and a co-morbid history of depression.

#### 2.2.2. Case 2

A 39-year-old divorced man, the father of two young children whom he shared the care with his ex-wife. He works full-time in the education field. Presented with a 20-year history of problematic gambling behaviours on sports betting both online and in the TAB. His gambling was a significant contributor to his depression with suicidal ideation. 

#### 2.2.3. Case 3

A 51-year-old married man with two teenage sons. His wife was unaware of the extent of his gambling which was a significant difficulty in their relationship. He lives in his own home and at assessment was an unemployed professional in the horticulture business. He presented with a 30-year history of gambling-related issues associated with online sports gambling and experienced co-morbid depression, low self-worth, and anxiety related to both his gambling and additional socioeconomic factors.

#### 2.2.4. Case 4

A 49-year-old married man living in the family home with his wife and two young children. His wife had been significantly affected by his gambling and could not offer any support for the client or his recovery. He was referred to the service for assessment after a significant suicide attempt related to his problematic gambling behaviours and alcoholism. He was significantly depressed and was experiencing multiple psychosocial stressors. However, at the commencement of treatment, he presented with no acute risks and had been abstinent from alcohol for several months so was suitable to commence the treatment program. 

### 2.3. Measures

*Problem Gambling Severity Index (PGSI)*: A nine-item scale for measuring the severity of gambling problems in the general population. The items are scored on a scale of 0–3, and responses are used to define four types of gamblers: non-problem (score = 0), low-risk (score = 1–3), moderate-risk (score = 3–7), and problem gamblers (score = 8+) [15]. 

*The Outcome Rating Scale* (ORS): A validated [16,17] online, self-completed rating of personal functioning and recovery across four domains of well-being—overall (general well-being), individual (personal well-being), interpersonal (family, close relationships), and social (work, friendships). Each domain is scored on a scale of 0–10, and the four domains are totalled to give a score out of 40, with higher scores indicating greater subjective well-being. Total ORS scores of <25 indicate clinical levels of distress.

*Problem and Goal Statements*: The aim of the problem and goal statements is for the client to describe as concisely as they are able what they perceive as their main problem and two specific and observable goals they wish to achieve in relation to the problem [18]. Participants then rate the extent to which the identified problem affects their daily activities from 0 (no interference) to 8 (severe interference). Participants also rate their current progress towards achieving the two goals on a scale from 0 (complete success) to 8 (no progress). Scores are totalled to give a score out of 24, with lower scores reflecting greater perceived progress toward resolving the problem.

*General feedback.* At the completion of the group, a brief, three-item feedback survey was administered to elicit qualitative data regarding the experience of attending the group, barriers, and enablers to participation, and most and least helpful elements of the group. 

The study was approved by the Alfred Health Human Research Ethics Committee.

### 2.4. Treatment

#### 2.4.1. Approach

Participants were treated using CBT with a specific focus on graded exposure and response prevention. Treatments such as graded exposure that address the “urge” to gamble are predominantly behavioural [19,20,21] and have shown to be a viable and effective treatment for problem gambling. This treatment approach is ideal for people experiencing overwhelming urges to gamble which often result in ongoing relapse and harm [22]. The aim of graded exposure is to gradually extinguish the clients’ urges to gamble through a stepwise progression [20] to directly break the two-way maintenance relationship between urges and gambling triggers such as money, gambling advertisements, sports events, boredom, stress, relationship problems and financial difficulties [21]. This process enables the client to achieve habituation of the gambling urge. Cognitive restructuring for negative thoughts related to depression and behavioural activation was also used in this group to supplement exposure therapy [23].

Treatment was conducted over a weekly evening online closed group program, facilitated by 2 specialist gambling service clinical staff members, and attended by all 4 participants during the first six months of treatment. Treatment was carried out in four main stages: (1) engagement, (2) graded exposure, (3) psychoeducation and (4) relapse prevention.

#### 2.4.2. Engagement

The initial part of the group treatment program focused on engaging the clients in the treatment process and for them to slowly acknowledge the consequences of their gambling. These clients have previously used gambling to avoid painful insights that caused them distress and maintained their gambling. To encourage group members to engage effectively they were supported to feel that they were part of the group and that their contributions to the group were acknowledged and valued [24]. The supportive group process enabled each participant to engage in a gradual process of identifying and learning from the harms of their gambling. Sharing their mutual experiences within the group setting was an important part of the first few group sessions that allowed the participants to obtain some relief from their negative emotional states and to accept responsibility for their gambling behaviours. Hope was enhanced by providing the participants with a theoretical understanding of how the treatment would work, and how it differed from counselling they had each attended over many years. 

Learning to experience emotional states and engage in self-observation was a slow process and proceeded according to the participant’s motivation to continue engaging in the exposure treatment by allowing the urges to subside and actively engaging in self-observation. This was done by the client slowly addressing the underlying emotional triggers for them to gamble, rather than escaping back to relapse. During this time, the gambler committed to engaging actively in the change process and believed recovery from their gambling addiction was possible. 

#### 2.4.3. Graded Exposure Program

The basic premise of the graded exposure treatment is that gambling behaviour is increased (reinforced) and maintained by the implicit excitement and rewards of winning and losing. This treatment program is based on behavioural therapy using cue exposure and response prevention for problem gamblers [20].

The exposure therapy used incorporated a stepwise grading process of online sports betting exposure tasks and response prevention. As part of the initial assessment and as treatment progressed, time was spent with each client to establish their usual gambling triggers that were graded from least triggering to most triggering stimuli as the client went through treatment. The weekly group consisted of working through a step-by-step graded exposure hierarchy using imaginal and in vivo exposure tasks to enable participants to habituate to tasks incrementally until their treatment goals were achieved. 

Further, therapists worked with participants prior to the commencement of treatment, to develop secure money management to ensure participants could not access money to gamble. Having money management in place allowed the participants to engage in the exposure tasks without the risks of actual gambling.

Graded exposure tasks included:Black and white and then colour pictures of online gambling sites and sports/racing events, made increasingly difficult as the client progressed through the hierarchy.Studying the races or sports games, anticipating which bet would be placed without placing a bet.Watching a sports game/horse race turning off the sport before the end without finding out the result.Picking the winning sports team or horse and watching the event, and then turning off the sport before the end without finding out the winner.

The grading of gambling cues allowed the participant to habituate to tasks one at a time until their end-of-treatment goal was achieved. Each new task usually takes 5–7 days of repeated daily exposure for habituation to occur. These tasks were performed at least five times a week. Each task usually lasts from 30 min to one hour. This time is reduced as habituation to the task was achieved and the participants’ urge was systematically extinguished. 

During each session, each client was provided an opportunity to discuss their tasks over the week and use their treatment diaries which were designed for them to record the strength of their urges before, during, and after treatment. 

#### 2.4.4. Relapse Prevention

Throughout treatment, relapse prevention was discussed to ensure each client understood the importance to feel confident to apply exposure techniques to all urges experienced after treatment using the same methods they have learned during the treatment. If the client experienced an urge to gamble after treatment completion, they were reminded to continue to allow these urges to subside as they have done repeatedly during treatment. After these urges pass, the client was encouraged to engage in their critical thought processes and challenge the erroneous beliefs that may have triggered the urge to gamble through cognitive therapy [25].

## 3. Results

All four participants completed 6 months of weekly exposure therapy in the online group. Case 1 was discharged following successfully completing 6 months of treatment and achieving his treatment goals and reaching a PGSI score of 0. Case 1 continued to participate in the group sessions once a month for three months to ensure his treatment gains were maintained and to learn new life skills described below. He also attended the 12 monthly follow-up appointments. Case 2 successfully completed treatment after eight months. At this time, he was involved in past legal issues unrelated to his gambling which became his focus and he withdrew from further follow-ups. Cases 3 and 4 continued to stay on for the group’s support following treatment completion as they were experiencing significant life/social stressors associated with COVID, loss of business, and relationship breakdowns. Despite these enduring life stressors, none of the clients had lapsed back into gambling at follow up sessions. The treatment in this phase was aimed at building the client’s resilience by equipping them with the knowledge to help them manage any ongoing co-morbid mental health concerns and improving their life skills such as assertiveness training, stress management, and scheduling new interests into their life now the participants were no longer gambling. 

### 3.1. PGSI Scores

Data were collected for all four participants at the commencement of the group and midway through the group, but only 3 of the four participants completed the PGSI after completion (Figure 1). At the commencement of the group, the mean score on the PGSI was 17.5 out of 27 (range 13–20) and all participants fell in the range of ‘problem gambling’. By the end of the program, the three participants who completed the measure had an average PGSI score of 2.7 (range 0–5). One of these fell into the non-problem gambler category, while the other two had lowered their scores to the moderate risk category. Case 3, who did not complete the PGSI at completion, scored a ‘0′ during the program, indicating he also met the criteria for ‘non-problem gambler.’ Paired samples t-tests were conducted for the three participants who completed all measures. Findings show that scores on the problem gambling severity index were significantly lower from pre to post intervention (*t*(2) = 10.193, *p* = 0.009) (Table 1).

### 3.2. Outcome Rating Scale (ORS) Scores

ORS scores were collected at multiple timepoints for all four participants. Case 1 started and ended with ORS scores in the “clinically significant” region (20/40), and despite some variability in his scores, they remained somewhat stable throughout treatment. All three other cases showed a “reliable improvement” (an increase of at least 5) in their ORS scores. Cases 2, 3, and 4 demonstrated “clinically significant change”, improving by at least 5 points and no longer scoring within the clinically significant range (Figure 2). 

### 3.3. Problems and Goals Scores

All four participants completed the problems and goals measured at commencement. Three of four rated the difficulty of their problem at 8 (most difficult), and one at 7. All four scored the achievability of their treatment goals at 8 (not achievable). At completion, the three participants who completed this measure scored the difficulty of their problem at a mean of 1.3 (range 0–3) (Figure 3), and rated the achievability of their treatment goals at an average of 0.8 (range 0–4) (Figure 4 and Figure 5). This improvement reached statistical significance for the compulsion to gamble problem (*t*(2) = 7.181, *p* = 0.019) and for both gambling reduction goals (Goal 1 *t*(2) = 5.00, *p* = 0.038. Goal 2 *t*(2) = 23.0, *p* = 0.002) (Table 2). At twelve months follow up, all three participants scored both the problem as 0 (achieved) and the two goals as 0 (easy).

### 3.4. Qualitative Feedback

The three participants who completed the program also took part in a follow-up feedback telephone call with a researcher independent of the program clinicians, to ascertain feedback regarding their experience. All three indicated that they enjoyed the group, despite some nervousness in the beginning: 


*At the start it was tricky, you’re getting to know the other guys and whatnot and we’re coming from very different backgrounds but yeah it certainly worked really well for me.*
(Case 3)

One of the things that was most valued was the shared experience with other men who had similar backgrounds:


*The shared experiences that we’ve had even though we didn’t know each other previously, so I guess, previously I’ve thought that it was just me. That no one else could be feeling like what I’m going through. So that’s one part of it and I guess helping the other guys as well. It seemed to help, you know, that we could all talk about it without judgement.*
(Case 2)

The use of the telehealth platform was also appreciated by participants, who found that it helped them to feel more secure: 


*Obviously, we talked about it, but the fact it was by video link, because it had to be because of covid, probably been a blessing in disguise because you didn’t have that sort of confrontational element of sitting in front of other patients I suppose. Which would have been ok in time I suppose but especially in the first few weeks it was nice to know that you were still in the safety and sanctity of your own home. And obviously you’re still sharing sensitive information, but it just seemed to work. Whether it was the dynamic of the group, but that platform worked well for everyone.*
(Case 3)


*That (videoconferencing) is probably more the winner as well because if you tell me to go to the hospital and do it there, or somewhere... but once you do it in the comfort of your own home or your car or wherever, you know, you get a little bit scared the first few times but doing it on my computer was a lot easier.*
(Case 1)

One participant summed up the importance of support in being able to make the change: 


*Oh, to be honest, it was probably my lifesaver. Whatever was offered to me I was going to take because I knew it was going to take a substantial effort from me but also the right resource to fix me long-term. So, when I started liaising with [the clinicians] and they offered me this program I just jumped at it. It was the best thing that could have happened to me. I think the biggest thing is the belief because I just had no belief that I could fix my gambling, and financially you just get to the point where think “Oh well financially, it doesn’t matter what happens, I’m not going to be able to jump over this”. But it’s just one foot in front of the other and it’s literally been a saving grace and allowed me to do things.*
(Case 3)

## 4. Discussion

The present study investigated the impacts of a CBT-informed exposure group therapy for individuals with gambling disorder using a case study design. Evaluation of exposure therapy elsewhere has shown it to improve gambling behaviour and decrease gambling-related distress [26], however its effectiveness when delivered online is not yet established. Ideally, clients may return to a normal lifestyle following participation in the group, without using modifying factors and avoidance strategies to eliminate the risk of gambling. The participants in our study showed significant improvement on standardised measures between commencing and completing their treatment. The clients who completed follow up remained free from any gambling at the 12-month follow-up and reported experiencing no urges or desire to gamble even in the presence of previously triggering materials and situations. Participants also reported their gambling had been resolved and that they had achieved their treatment goals. Before beginning the program, participants had experienced significant impacts on their relationships because of their gambling and had contemplated suicide as an option. The current program provided a supportive space where the participants could connect with each other and where they could experience a sense of hope for recovery. These factors were considered by each participant as important for their commitment to the therapy. They also felt comfortable in the online space to share their stories and move forward together in their recovery feeling a sense of belonging and comradery.

When clients completed the PGSI measure after completing the group treatment program, the facilitators noted that participants had difficulty with questions 5 to 9 on the 9-item PGSI [27]. Despite reporting having not gambled over the last 12 months and experiencing no urges to gamble, these clients experienced significant guilt concerning their past gambling behaviour and the associated harms across a range of life domains [2,28,29,30,31]. It was evident these gamblers experienced much regret [32] as they achieved recovery and realised or imagined how their present situation and family life would have been better had they not gambled.

As Hing et al. [33] noted, stigmatisation of the problem gambling population is harsh, the general public often views gambling harm as an issue of willpower, and negative self-stigma is even greater. Participants of the group self-identified as problem gamblers, a self-identity they found hard to shift. It is therefore understandable that Question 5 of the PGSI, “Have, you felt that you might have a problem with Gambling?” and Question 7, “Have you ever felt guilty about the way you gamble or what happens when you gamble?” could cause some distress for the gambler in recovery. This finding highlight that despite completing treatment and not gambling at long-term follow-up, participants would still be categorised as moderate-risk gamblers on the screening tool, despite qualitatively advising they were no longer at risk. All discussed how they still experienced significant guilt associated with their past gambling. 

It is evident in the study that individuals who complete treatment and no longer desire to gamble develop must have support to deal with the traumas associated with the losses they have to endure because of past problem gambling behaviours. Therefore, a missing component of gambling treatment appears to be the provision of evidence-based therapies to help address the traumas of past gambling harms. Understanding how these past gambling harms impact the gambler as they enter recovery is crucial. 

The present results support previous studies finding that exposure therapy can help clients to recover from their gambling-related problems and no longer experience any urges to engage in any form of betting including sports betting [26], and that this can be done through an online platform [21]. The group being run online did not have an adverse impact on its effectiveness and positive therapeutic factors such as the therapeutic alliance did not appear to be impacted [11]. In fact, participants felt that being able to attend the program without leaving their homes was a great strength of the group. 

For these gamblers, exiting the relapse cycle was enhanced with a supportive group environment provided by peers and trained clinicians. Throughout the treatment, participants’ contributions to the group were acknowledged and valued [24] and they were provided with the hope that recovery could be possible [34]. This supportive process gradually allowed the client to face and effectively deal with the consequences of their gambling which for many had been overwhelming, and in the past had led them to ongoing relapse [35]. 

This innovative program provides encouraging preliminary evidence for the provision of treatments to clients with gambling disorders via Telemedicine. Accessibility to this program was of specific relevance during the COVID-19 lockdowns as these participants were able to gain access to weekly support and evidence-based treatments from their own homes. Treatment success was measured using several domains over the client’s life. For example, relating to the gambling urge, and quality of life which demonstrated clients were gambling-free and experienced an increased sense of self-efficacy over their past gambling behaviours and a more fulfilled life. 

### Limitations

A limitation of the present study was the small sample size and, as such, quantitative data needs to be interpreted with caution. However, coupled with the qualitative data, the present study provides some promising preliminary results suggesting that a formal randomised controlled experimental design is warranted to test the efficacy of this online group treatment.

## 5. Conclusions

This real-world, evaluation study has provided initial evidence suggesting that online group exposure therapy can be effective in helping people affected by gambling harm to engage in treatment and successfully complete a therapeutic program. While the number of participants is small, the fact that all four were able to reduce their gambling behaviour through participating in this group after many years of unsuccessful interventions elsewhere provides preliminary evidence that the group was effective. Standardised scores for three of the four participants indicate significant improvement in both problem gambling behaviours and in their individual problem and goal attainment. The flexibility of an online treatment program using a group format was also acceptable to participants and may be an option to engage others who have barriers to accessing evidence-based treatments in person. Typical barriers to engagement may include shame, embarrassment, stigma; unwillingness to admit a problem; geographic factors such as rural and remote location; logistics of transport and timing; social anxiety about attending a group. The online modality allowed for the provision of services that meet the needs of the participants, rather than the convenience of the therapists or the healthcare service. These findings suggest further research in this field is warranted, expanding the range of participants to include a wider demographic profile and a randomized control study with long-term follow-up across a number of the clients’ life domains and gambling behaviours.

## Figures and Tables

**Figure 1 ijerph-19-16273-f001:**
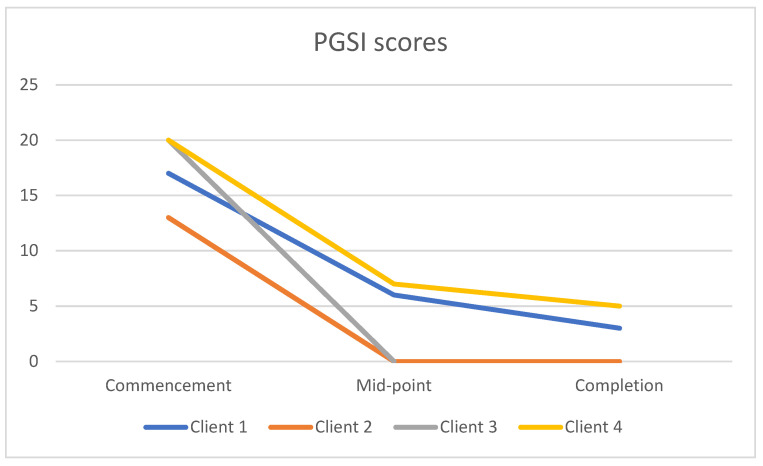
Change in Problem Gambling Severity Index (PGSI) scores during treatment.

**Figure 2 ijerph-19-16273-f002:**
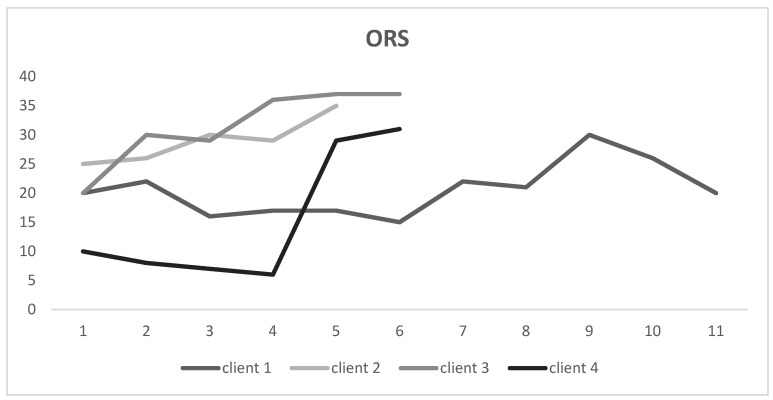
Change in Outcome Rating Scale (ORS) scores during treatment.

**Figure 3 ijerph-19-16273-f003:**
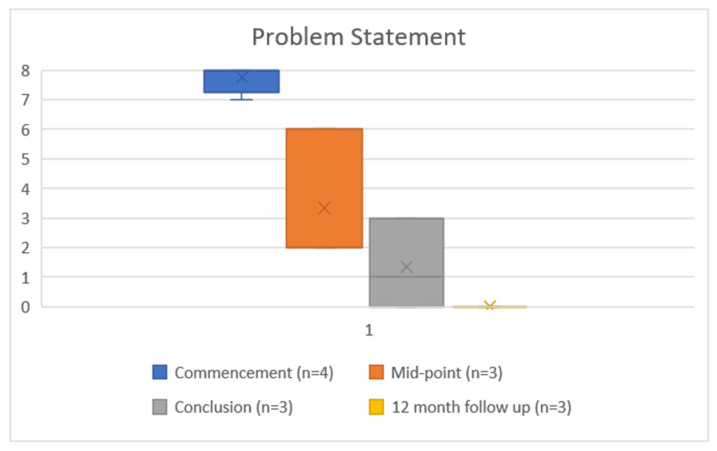
Problem Statement, where 0 = easiest problem to manage, 8 = most difficult problem to manage.

**Figure 4 ijerph-19-16273-f004:**
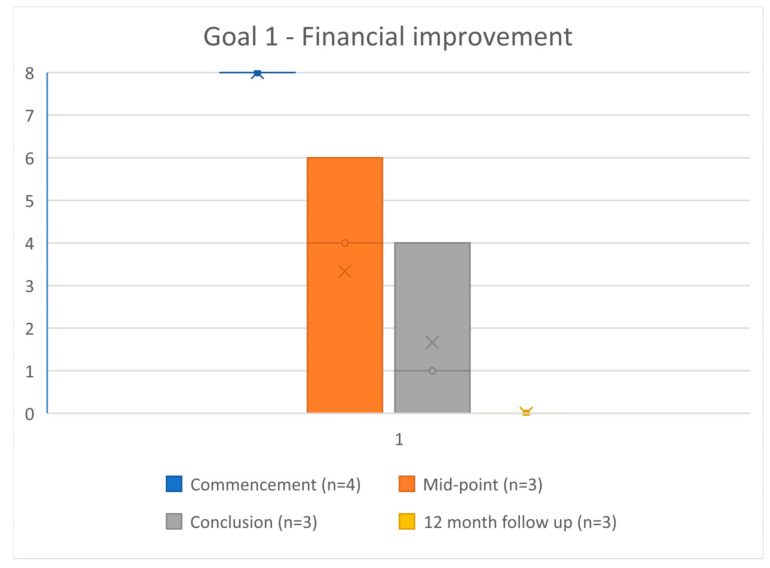
Goal 1—Financial improvement, where 0 = goal is achievable, 8 = goal is not achievable.

**Figure 5 ijerph-19-16273-f005:**
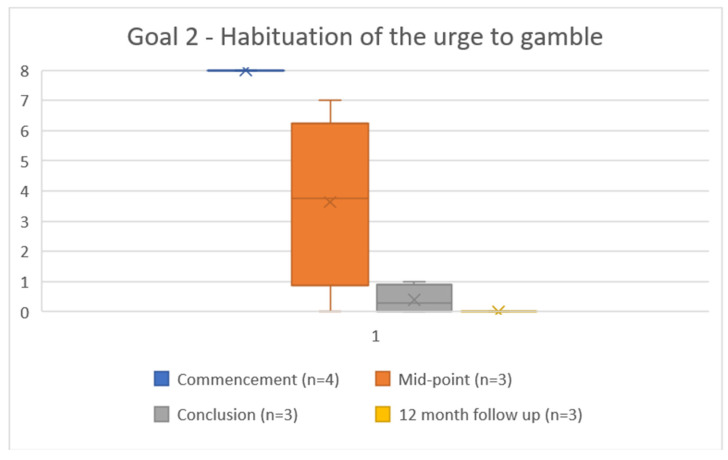
Goal 2—Habituation of the urge to gamble, where 0 = goal is achievable, 8 = goal is not achievable.

**Table 1 ijerph-19-16273-t001:** PGSI paired samples test—Commencement to Completion.

Mean	Std. Deviation	Std. Error Mean	95% Confidence Interval of the Difference		t	df	Sig. (2-Tailed)
			Lower	Upper			
20.66667	3.51188	2.02759	11.94266	29.39067	10.193	2	0.009

**Table 2 ijerph-19-16273-t002:** Problem and Goals Statements paired samples test—Commencement to Completion.

	Mean	Std. Deviation	Std. Error Mean	95% Confidence Interval of the Difference		t	df	
				Lower	Upper			Sig. (2-Tailed)
Problem Commence—Problem Completion	6.33333	1.52753	0.88192	2.53875	10.12792	7.181	2	0.019
Goal 1 Commence—Goal 1 Completion	6.66667	2.30940	1.33333	0.92980	12.40354	5.000	2	0.038
Goal 2 Commence—Goal 2 Completion	7.66667	0.57735	0.33333	6.23245	9.10088	23.000	2	0.002

## Data Availability

The data presented in this study are available on request from the corresponding author. The data are not publicly available due to privacy concerns with small numbers of participants.

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
