# Peer review of "Case Study Demonstration of the Potential Acceptability and Effectiveness of a Novel Telehealth Treatment for People Experiencing Gambling Harm"

_ijerph, 2022, doi:10.3390/ijerph192316273_

Round 1
Reviewer 1 Report
Dear authors, it’s an interesting article but there are a few things.
first of all you noticed the following in your limitation part:
“the present study provides some promising preliminary results suggesting that a formal randomised controlled experimental design is warranted to test the efficacy of this online group treatment.” please also add that to your abstract but also to your conclusion.
The other limitation which you don’t mention is that there is no control group and you also need to add that to your abstract but also to your conclusion. That way readers will understand after reading your abstract / conclusion, that it is a small study without a control group. In other words that is essential information which should be in the abstract and conclusion. But the extra limitations also need to be mentioned under limitations. The same applies for the fact that your study only included people in their 40s so not young people, and only guys. Consequently, you need to amend the title of your study because your conclusion only applies to guys because there were no women in your study.
Line 285 you state that three participants completed the program yet three continued to stay in the program so you need to clarify that.
“The participants referred to the program were four Australian males aged between 45 and 51 years.” You need to change that because according to your own study, case 2 was 39.
Your study is about online betting but for case 4, there is nothing about (online) betting, so what is his betting problem?
Also, you need to add something about the fact that even though you say your treatment is effective, in only one of the four it was effective to the point that he did not need any help anymore; 3 of the four patients remained in the program to continue treatment. So then you need to question the fact if the treatment is really effective or not. So please address that in the discussion part.
Reviewer 2 Report
This manuscript provided evidence about the effectiveness of telehealth treatment for gambling disorders. Four online gamblers met the criteria for gambling disorder were recruited. After completing the treatment, participants no longer met the criteria for a gambling disorder.
The authors provided a detailed review of the literature related to the topic. The information related to participants, measures, and treatment was clearly described. Based on the provided quantitative and qualitative results, I think it is reasonable to hypothesize that telehealth treatment is effective.
My primary concerns about the manuscript include 1) In the experiment design, there is no control group. The scores may improve, as described in Figures 1-3, even if no treatment is provided. When there is no control group, it is impossible to identify the causal relationship between the treatment and the improvement. 2) The authors do not provide statistical evidence related to the improvement of the scores. Although the data demonstrate an improving trend in the score, it is unclear whether it is due to a random effect or it is statistically significant. 3) The sample size is very small (only four) and might not be representative. Based on the samples, it is insufficient to conclude that the treatment is effective in general.
In summary, I believe the evidence described in this manuscript is insufficient to demonstrate the effectiveness of telehealth treatment due to the limitations I described in the previous paragraph. I do not suggest the acceptance of this manuscript.
Reviewer 3 Report
This study might be best suited to be published as a case report instead of a research article. Please follow the journal guidelines for case report writing, such as this one:
https://www.mdpi.com/1660-4601/18/4/2083
For a research article, the authors may need to consider adding more details about statistical analysis for pre-/post-test experimental designs.
Round 2
Reviewer 1 Report
Dear authors, thank you for making the changes.
Author Response
Thank you.
Reviewer 2 Report
Thanks for the clarification. I suggest the authors perform statistical tests, even if no statistical significance is observed due to the small sample size. In some measures, such as PGSI and Problems & Goals scores, the mean post-treatment value might be significantly lower than the mean pre-treatment value.
The caption of the plot of the Problems and Goals scores is missing.
In Figures 3,4, and 5, "Conclusioon" is a typo.
Do "pre," "midway," and "post" in Figure 1 correspond to "Commencement," "Midway," and "Conclusion" in Figures 3,4, and 5, respectively?
Do time points 1, 2, and 3 in the plot of Problems and Goals scores correspond to "Commencement," "Midway," and "Conclusion" in Figures 3,4, and 5, respectively?
